# Ecologically Valid Explanations for Label Variation in NLI

**Nan-Jiang Jiang**[*]
Department of Linguistics
The Ohio State University
jiang.1879@osu.edu

**Chenhao Tan**
Department of Computer Science
The University of Chicago
chenhao@uchicago.edu

**Marie-Catherine de Marneffe**
FNRS, UCLouvain
marie-catherine.demarneffe
@uclouvain.be

## Abstract

Human label variation exists in many natural language processing (NLP) tasks, including natural language inference (NLI). To gain direct evidence of how NLI label variation arises, we build LIVENLI, an English dataset of 1,415 ecologically valid explanations (annotators explain the NLI labels they chose) for 122 MNLI (Williams et al., 2018) items (at least 10 explanations per item). The LIVENLI explanations confirm that people can systematically vary on their interpretation and highlight *within-label variation*: annotators sometimes choose the same label for different reasons. This suggests that explanations are crucial for navigating label interpretations in general. We few-shot prompt language models (LMs) to generate explanations but the results are inconsistent: the models sometimes produce valid and informative explanations, but they also generate implausible ones that do not support the label, highlighting directions for improvement.

## 1 Introduction

Practices for operationalizing annotations in NLP datasets have largely assumed one single label per item. However, human label variation (Plank, 2022) has been found in a wide range of NLP tasks (Plank et al., 2014; Poesio et al., 2018; Pavlick and Kwiatkowski, 2019; Nie et al., 2020; Uma et al., 2021; Jiang and de Marneffe, 2022, i.a.). Aroyo and Welty (2015), i.a., argued that variation that is systematic in annotations should be considered signal, not noise. Specifically, the NLI task – identifying whether a hypothesis is true (Entailment), false (Contradiction), or neither (Neutral) given a premise – has embraced label variation and set out to predict it (Zhang et al., 2021; Zhou et al., 2022). However, it remains an open question where label variation in NLI stems from.

We introduce the LIVENLI[1] dataset, containing 122 re-annotated MNLI (Williams et al., 2018) items, each with at least 10 highlights and free-text explanations for the labels chosen by the annotators, totaling 1,415 explanations. The dataset is relatively small because we intended it to be suitable for few-shot evaluation, following recent trends in BIG-Bench authors (2023).

The contribution of LIVENLI's explanations is their ecological validity: annotators provide both the label and the explanations, in contrast to many explanation datasets (e.g., e-SNLI (Camburu et al., 2018)) in which explanations were collected for a *given* "ground truth" label. LIVENLI hence provides direct evidence for how label variation among annotators arises. In addition, LIVENLI highlights another kind of variation, *within-label variation*: in some cases, humans converge to the same label for different reasons. Finally, we evaluate whether GPT-3 (Brown et al., 2020) can generate explanations for label variation. We found that its explanations can be fluent and informative but also implausible or not supporting the label.

## 2 Data Collection

We re-annotated 122 items from the MNLI dev set where each item originally had 5 annotations:
- 60 items where only 2 annotators agreed on the label in the original MNLI annotations;
- 50 items where 3 annotators agreed on the label in the original MNLI annotations, reannotated in Nie et al. (2020)'s ChaosNLI with 100 annotations;
- 12 items (4 for each NLI label) where all 5 annotators agreed on the label in the original MNLI annotations.

The first 110 items were analyzed by Jiang and de Marneffe (2022) who proposed a taxonomy of reasons for label variation.

---

[*]Now at Google.

[1]Label Variation and Explanation in NLI, available at https://github.com/njjiang/LiveNLI.

Ex.1 **P:** Most pundits side with bushy-headed George Stephanopoulos (This Week), arguing that only air strikes would be politically palatable. **H:** Mr. Stephanopoulos has a very large pundit following due to his stance on air strikes only being politically palatable. LiveNLI [E,N,C]: [0.4, 0.3, 0.3]

QUD: Does Stephanopoulos have a very large pundit following?
Resp.1.1: E – This statement is most likely to be true because in the context is stated that "Most pundits" would side with Mr. Stephanopoulos. Most pundits could also mean a very large pundit following.
Resp.1.2: N – You cannot infer that the overall number of pundits following the individual is large just because the majority of pundits follow the individual. He could just have 2 out of 3 total pundits following him, for instance. Furthermore, they may be following him for reasons outside his stance on air strikes.
QUD: Do pundits follow Stephanopoulos due to his stance on air strikes?
Resp.1.3: N – George Stephanopoulos may have a follow from pundits, but it might not be due to his support of drones.
Resp.1.4: C – He might have a large pundit following, but that would have to be for something before the current issue of air strikes since one event wouldn't get people a large following overnight.

Ex.2 **P:** for a change i i got i get sick of winter just looking everything so dead i hate that
**H:** I'm so sick of summer. LiveNLI [E,N,C]: [0, 0.35, 0.65]
Resp.2.1: C – The context is stating how one is sick of winter, not summer, as the statement describes.
Resp.2.2: C – The speaker hates winter because the foliage is dead, therefore he likely loves summer when everything is alive.
Resp.2.3: N – The context mentions being sick of winter while the statement mentions being sick of summer. These could both be true because the same person may still complain of summer's heat.

Table 1: Examples in LiveNLI. **P**: Premise. **H**: Hypothesis. [E,N,C]: the probability distributions over the labels (E)ntailment, (N)eutral, and (C)ontradiction, aggregated from the multilabel annotations in LiveNLI.

**Procedure** Annotators read the premise (context) and hypothesis (statement), and were asked whether the statement is most likely to be true/false/either true or false (corresponding to Entailment/Contradiction/Neutral, respectively). Multiple labels were allowed. They were then asked to write a free-text explanation for all the label(s) they chose. Inspired by Camburu et al. (2020); Wiegreffe et al. (2022); Tan (2022), we asked annotators to give explanations that provide new information and refer to specific parts of the sentences, and to avoid simply repeating the sentences. Annotators were also asked to highlight words from the premise/hypothesis most relevant for their explanations. 48 native speakers of English were recruited from Surge AI (`surgehq.ai`, details in A1).

## 3 Analysis of LiveNLI

**Label descriptive statistics** We normalize each multilabel response into a distribution, then average the individual distributions to obtain the item's label distribution. Figure 1, where each set of stacked bars gives the distributions for one item, shows that label variation is widespread, even on the 12 items that received full agreement in MNLI (right panel). The annotators provided legitimate explanations for the labels that differ from the original one (see (1) in A1), suggesting that the variation is genuine and not noise. This reiterates Baan et al. (2022)'s point that considering one distribution to govern all judgments from the population is a simplification, and distributions can shift with annotator pools.

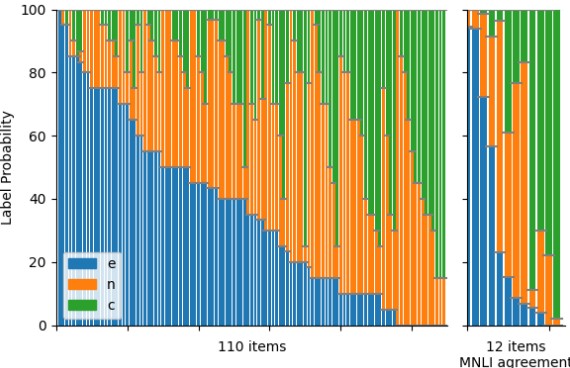

Figure 1: Average distributions from the normalized multilabel responses for each item.

**Annotators' reasons for label variation** Jiang and de Marneffe (2022) analyzed the reasons for NLI label variation from a linguists' perspective and built a taxonomy of linguistic phenomena that can lead to different interpretations. For instance, in Ex.1 (Table 1), does one infer *a large following* in the hypothesis from *most* in the premise? They hypothesized that such lexical indeterminacy contributes to the variation reflected in the NLI label distribution. However, their taxonomy was built post-hoc, detached from the annotators. Here we leverage LiveNLI's ecological validity: we analyze the reasons for label variation from the perspective of linguistically-naive annotators.

For each pair of premise/hypothesis, one author assigned it one or more taxonomy categories exhibited in its explanations (Table 5 in A2 shows the frequency of the categories). Across the board, label

variation arises in similar reasons to what Jiang and de Marneffe (2022) hypothesized. We calculated agreement between our taxonomy reannotations and the annotations from Jiang and de Marneffe (2022). The Krippendorff's $\alpha$ with MASI-distance is 0.30, suggesting low agreement (contingency matrix in Figure 3 in A2). Specifically, the average number of categories used here is higher – 1.33 per item vs. 1.22 in Jiang and de Marneffe (2022), suggesting nontrivial differences between the annotators' and linguists' reasons for label variation.

Furthermore, the explanations uncover more ways in which label variation can occur. In particular, we found that label variation often stems from annotators judging the truth of different at-issue content, which answers different Questions Under Discussion (QUDs) (Roberts, 2012). For example, in Ex.1, Responses 1.1 and (the first half of) 1.2 take the main point of the hypothesis to be Stephanopoulos having a very large pundit following, but have different judgments on whether this main point is true. On the other hand, 1.3 and 1.4 focus on the reason for which pundits follow Stephanopoulos, and agree with 1.1 that he has a large following. We thus added a "QUD" category to the taxonomy.[2] The QUD category occurred in 28 items (out of 122) and is the third most frequently used category, thus an important aspect of language understanding. Incorporating QUD into NLP tasks and modeling (De Kuthy et al., 2018; Narayan et al., 2022) is an interesting direction.

**Within-label variation** Our analysis reveals that there is also *within-label variation*: annotators may vary in their text understanding when giving the same label. There are 16 instances of within-label variation: annotators chose the same labels but provided very different explanations.[3] As mentioned earlier, 1.2 and 1.3 both chose Neutral but assumed different QUDs (1.2 also encompasses 1.3).

As another example, in Ex.2, the explanations indicate that the annotators differ in whether to take the premise and hypothesis to be referring to the same entity/event. Response 2.1 assumes that the premise and hypothesis refer to the same season: since the premise talks about winter and not summer, the hypothesis *I'm so sick of summer*

is false. 2.2, on the other hand, does not assume that the premise and the hypothesis refer to the same season, and infers through probabilistic enrichment that the speaker likes summer, making the hypothesis false. Thus both with and without the coreference assumption, annotators label the item as Contradiction.

Within-label variation shows up in ∼13% of LIVENLI, and highlights the diversity of human explanations. This suggests that model generated explanations should be similarly diverse to imitate human decision-making.

## 4 Predict and Explain Label Variation

Models should capture inherent label variation to achieve robust understanding. Explanations are essential for navigating the uncertainty rising from such variation. Here we use LIVENLI to few-shot prompt LMs to jointly predict and explain label variation. We used text-davinci-003, a variant of GPT-3 (Brown et al., 2020) with few-shot prompting (example in Table 7).[4] Label variation in NLI has been predicted either as a distribution over three labels (Zhang et al., 2021; Zhou et al., 2022) or a multilabel (Jiang and de Marneffe, 2022; Liu et al., 2023). Here we use the multilabel approach for its greater interpretability compared to distributions. We select as gold answers labels with probability (from LIVENLI or ChaosNLI) above 0.2, similar to Jiang and de Marneffe (2022).

We use Predict-then-Explain (P-E) and Explain-then-Predict (E-P), a.k.a. chain-of-thought prompting (Wei et al., 2022), which has been shown to improve performance of reasoning tasks like NLI (Ye and Durrett, 2022). (P)redict-only prompts the model without explanations for the same items, giving a baseline measuring the effect of explanations.

**Train/test items** Since Jiang and de Marneffe (2022) found that models are sensitive to label imbalance, we used a label-balanced set of 11 training items from LIVENLI, with one item per single-label and two items per multiple-label. The test set includes the rest of LIVENLI, augmented with ChaosNLI for infrequent labels, totaling 259 items.

**Metrics** The metrics include exact-match accuracy and macro F1. To measure how well the model can separate the agreement items from ones with label variation, we calculate a 4way-F1 score by

---

[2]It replaces two of Jiang and de Marneffe (2022)'s categories, "accommodating minimally added content" and "high overlap", which involve annotators having different readings and ignoring certain parts of the items.

[3]We consider two explanations to be different if they focus on different sentence parts or make different assumptions.

[4]We did not receive access to GPT-4 and ChatGPT performed worse in preliminary experiments.

| | Accuracy | F1 | 4way F1 | # labels |
|---|---|---|---|---|
| P | $25.25_{1.91}$ | $\mathbf{70.70}_{1.70}$ | $37.05_{0.92}$ | $1.92_{0.35}$ |
| P-E | $24.7_{3.25}$ | $69.00_{3.68}$ | $\mathbf{37.15}_{1.77}$ | $1.78_{0.32}$ |
| E-P | $19.90_{0.28}$ | $65.85_{4.46}$ | $27.55_{1.48}$ | $1.90_{0.28}$ |
| ENC | 14.30 | 72.70 | 18.20 | 3.00 |

Table 2: Metrics for each model and mean number (#) of predicted labels. The average number of gold labels in the test split is 1.71. Each value is the average of two random splits (standard deviation in subscript).

| | Validity | | Informativeness | | |
|---|---|---|---|---|---|
| | Plausible | Support | Specific | New Info | # Total |
| E-P | 75.0 | 80.9 | 38.2 | 10.3 | 68 |
| P-E | 77.1 | 81.4 | 37.1 | 5.7 | 70 |

Table 3: Percentage of explanations (out of # Total) for 35 items marked as Plausible, Support label, Specific, and introduce New Info.

calculating macro F1 scores on a 4way categorization of the multilabels: three single labels vs. a "complicated" label if there are multiple labels.

**Classification Results** Table 2 shows the performance of each model, together with the uniform baseline of always predicting all three labels ENC.[5] Both P and P-E significantly outperforms E-P.[6] This adds to the series of results where having post-prediction explanations outperforms chain-of-thought style prompting (Lampinen et al., 2022; Zhou et al., 2023). In particular, P-E predicts fewer labels than E-P, suggesting that the models consider more labels to be possible when generating explanations first. P predicts the most number of labels, which can potentially be due to the fact that it does not need to generate an explanation for each label predicted, and hence does not pay the long sequence penalty when generating more labels.

**Explanations Results** We performed a qualitative analysis to evaluate the models' explanations on four criteria inspired by Wiegreffe et al. (2022): do the explanations plausibly describe sentence meaning (Plausible), support the label (Support), go beyond restating the sentences (Specific), introduce new information about the subject matter (New Info) (details in A4). The first two criteria target the explanations' validity, and the last two

---

[5]The ENC baseline scores high on F1 because it has a perfect recall, and a high precision given that more than half of the items are multi-labeled.

[6]$p = 0.02$ comparing P vs. E-P, $p = 0.03$ for P-E vs. E-P, $p = 0.84$ for P vs. P-E, McNemar's test.

---

Ex. 1 **P:** In Roman times a temple to Jupiter stood here, followed in the fourth century by the first Christian church, Saint-Etienne. **H:** Saint-Etienne, a Christian church, had a temple to Jupiter and performed rituals during Roman times
ChaosNLI [E,N,C]: [0.25, 0.25, 0.50]
**Generic**, **Plausible**, **Not Support**, **No new info**
**P-E:** C – the context states that there was a temple to Jupiter before the church was established, while the statement implies the church had a temple and performed rituals. This may or may not have been true, but it is not something that is stated in the context.
**Specific**, **Plausible**, **Support**, **New info**
**E-P:** N – it is possible that a church in Roman times may have done rituals for Jupiter, as Christianity was still a nascent religion at that time, but it is unknown based on the context.

Ex. 2 **P:** Most pundits side with bushy-headed George Stephanopoulos (This Week), arguing that only air strikes would be politically palatable. **H:** Mr. Stephanopoulos has a very large pundit following due to his stance on air strikes only being politically palatable.
LIVENLI [E,N,C]: [0.4, 0.3, 0.3]
**Generic**, **Plausible**, **Support**, **No new info**
**P-E:** N – the context implies a large following but does not specify an exact amount. Therefore it cannot be definitively stated if there is a large following or not.
**Generic**, **Implausible**, **Support**, **No new info**
**E-P:** C – is is unknown to what extent the pundit's following is and isn't large. It has only been alluded to that many pundits are agreeing with him.

Table 4: Generated explanations from both models.

their informativeness (Leake, 1991).

For a random sample of 35 items, one author provided binary judgments on the four criteria for both models' explanations, with the model identity hidden. As shown in Table 3, the models do not differ much in their explanations quality, except for New Info. ~80% of the explanations support the labels they predict. Those not supporting the label are often Contradiction explanations (across models, 67.8% Support for Contradiction vs. 88.2% for other labels). In Ex. 1 in Table 4, the explanation only states that the statement is not definitely true (hence not Entailment) but fails to state why it is false, thus not supporting the Contradiction label. Furthermore, only ~60% of the explanations are both plausible and support the label (hence valid).

The explanations are also lacking informativeness: over half of the explanations are merely restating the premise/hypothesis (not Specific) and rarely have new information. However, when the generated explanations are both valid and informative (as in the Neutral one for Ex. 1 in Table 4), they do contribute to further understanding of why the label applies by providing interpretation.

Moreover, the generated explanations do not exhibit the range of within-label variation found in

the human explanations in Table 1. For Ex. 2 in Table 4, both generated explanations focus on the aspect of whether "most" implies "large", while the human explanations in Table 1 identify two different QUDs the sentences can address.

## 5   Conclusion and Discussion

We built LIVENLI providing ecologically valid explanations for NLI items with label variation. We showed that LIVENLI's explanations help us understand where label variation stems from, and identify Question Under Discussion as an additional source of label variation. We further emphasize a deeper level of variation: within-label variation. Our work illustrates the utility of explanations and provides future avenues for improving model capability of leveraging and generating explanations.

## Limitations

The LIVENLI items comes from the MNLI dataset, which is known to contain annotation artifacts and biases (Gururangan et al., 2018; Geva et al., 2019). It is possible that those biases also contribute to the variation found here. Further investigations on other datasets or tasks are needed to better understand the prevalence of the label variation and within-label variation.

As shown in A1, filtering responses by answers to control items – the usual technique for quality control and reducing noise – no longer works here. It is possible that using a different set of control items, perhaps hand-written instead of sourced from an existing corpus, would yield different results. In any case, the most effective way to reduce noise is to control quality of the workers, which is what Nie et al. (2020) and we did. Still, noise is inevitable in any data collection process, and we did find some amount of noise in LIVENLI: there are 6 responses from 3 items where the explanations reveal that the annotators misread the sentences. Those 6 responses have labels that differ from the rest of the responses for those 3 items. Therefore, a small amount of label variation found here is contributed by noise.

Our dataset is English-only. Studying label variation in other languages may reveal new linguistic phenomena contributing to variation that are not seen in English, and further reshape the taxonomy of reasons for label variation.

Our evaluation of the explanations generated are on a small sample and is performed by one au-

thor. An evaluation scheme with multiple annotators would provide a better assessment of the model quality. We only tested one model: text-davinci-003, because it was the best model available to us at the time of development. We have not received approval for access to GPT-4 from OpenAI. Furthermore, due to the closed-nature of many commercial language models, including text-davinci-003, we do not know what their training data contains. It is possible that the training set for text-davinci-003 includes the items from MNLI (where the LIVENLI items are sourced from) and therefore contributed to some of the advantages of the models shown here.

## Broader Impact

We showed that GPT-3 can generate explanations that are detailed and closely resembling the structure of human explanations. However, the explanations may not be as grounded in the input premise/hypothesis and the labels they predict. This can lead to potential misuse – the explanations can be used to manipulate humans into believing the conclusion/label being explained, especially when the explanations are specific and detailed, giving the impression of correctness and authority. Therefore, these models should not be deployed without further studying their social implications.

## Acknowledgments

We thank Michael White, Micha Elsner, the members of the Clippers group at OSU, the C.Psyd group at Cornell, and the CHAI group at UChicago for their insightful feedback. We thank the workers at Surge AI who provided valuable data that enabled this work. This material is based upon work supported by the National Science Foundation under grant no. IIS-1845122. Marie-Catherine de Marneffe is a Research Associate of the Fonds de la Recherche Scientifique – FNRS.

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

# Appendix

## A1 Details on Data Collection Procedure

Annotators were paid $0.6 per item (hourly wage: $12 to $18). Each item has at least 10 annotations. On average, each annotator labeled 29.3 items. Our data collection was performed with IRB approval. SurgeAI guarantees that the annotators are native English speakers and have passed certain reading and writing tests. Figure 2 shows a screenshot of the annotation interface.

(1) shows an example where the original MNLI annotations are unanimously Neutral but exhibiting genuine variation in LIVeNLI annotations, together with the LIVeNLI explanations.

(1) **P:** The original wax models of the river gods are on display in the Civic Museum. **H:** Thousands of people come to see the wax models.
original MNLI: [0,1,0]
LIVeNLI [E,N,C]: [.23, .73, .04]
N – The context refers to the wax model displays in the museum. The context makes no mention of the number of visitors mentioned in the statement.
E – Museums are generally places where many people come, so if the original wax models are there it is likely thousands of people will come to see them.
Entailment | Contradiction – It's unlikely a museum could stay open for very long without thousands of visitors, so it's likely true that thousands of people come to see these wax mdoels. Unless, of course, it's a big museum with many attractions more interesting than the models, in which case the statement is likely to be false.

## A2 Details on Taxonomy Reannotations

Table 5 shows the frequency of taxonomy reannotations. Figure 3 compares our reannotations with Jiang and de Marneffe (2022)'s annotations.

**Read the following context and statement:**

**Context:** Could you please speak to this issue, with regard to the social ramifications of gum chewing in public?

**Statement:** You don't have an opinion on gum chewing in public, I see.

**Choose one or more from the following:**

If you feel uncertain and you feel that multiple options apply, choose them all instead, even though it might feel contradictory.

**Assuming the context is true, the statement:**

☐ is most likely to be true

☐ can be either true or false

☐ is most likely to be false

**Explain, in a few sentences, why you chose your answer.**

If you chose more than one option, elaborate in which circumstances each option is possible.

**Explain all the options you chose.**

Your explanation should include **new information** and **refer to specific parts of the sentences**. It should **NOT simply repeat the sentences**

Avoid "The context and statement means the same/opposite thing". **Specify which part of the context and statement means the same/opposite thing.**

Avoid "Just because X doesn't mean Y". **Say under what circumstances X does not mean Y, or say that X can mean Y or Z.**

Avoid "The statement is ambiguous/it's not clear what it means". **Elaborate what the possible meanings are and why it is ambiguous.**

Minimum word count: 10  Words: 0

**Highlight the words in the Context and Statement that are relevant to your explanations.**

Your explanations should refer to specific words/parts of the sentences. Highlight those words and phrases that your explanations mentioned.

**Only highlight the words that are most important for the explanations.**

Figure 2: Screenshot of the data collection interface.

| Combination of Taxonomy Categories | Frequency |
|---|---|
| Probabilistic | 28 |
| Lexical | 21 |
| *QUD* | 15 |
| Coreference | 6 |
| Lexical | Probabilistic | 5 |
| Coreference | Probabilistic | 5 |
| Imperfection | Lexical | 5 |
| Implicature | 5 |
| Imperfection | Probabilistic | 4 |
| Probabilistic | *QUD* | 4 |
| Coreference | Lexical | 3 |
| Lexical | *QUD* | 3 |
| Temporal | 2 |

Appear once:
Implicature | Interrogative; Implicature | Lexical; Interrogative; Implicature | *QUD*; Imperfection | *QUD*; Interrogative | Lexical; Coreference | Temporal; Coreference | Imperfection; Coreference | *QUD*; Implicature | Interrogative; Implicature | Lexical; Interrogative; Implicature | Lexical | Probabilistic | *QUD*; Probabilistic | Temporal; Presupposition | *QUD*; *QUD* | Temporal

Table 5: Frequency of the taxonomy categories in LIVeNLI. Label variation in annotators arises for similar reasons as what Jiang and de Marneffe (2022) hypothesized. QUD is a new category.

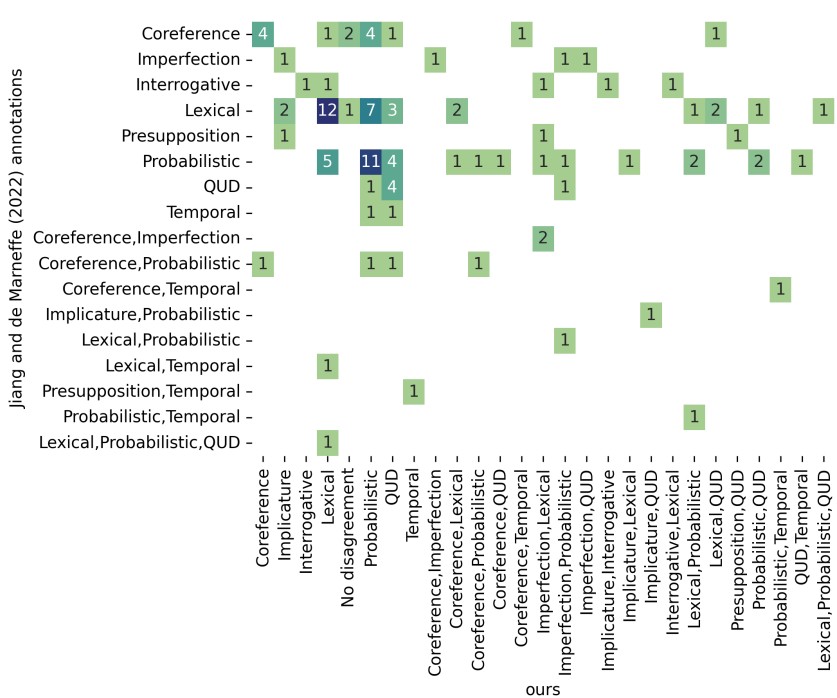

Figure 3: Contingency matrix of current taxonomy reannotations vs. Jiang and de Marneffe (2022)'s annotations.

## A3 Details on Prompting Setups

| | train LIVENLI | test total | LIVENLI | ChaosNLI |
|---|---|---|---|---|
| E | 1 | 37 | 18 | 19 |
| N | 1 | 37 | 13 | 24 |
| C | 1 | 37 | 8 | 29 |
| NC | 2 | 37 | 26 | 11 |
| EC | 2 | 37 | 1 | 36 |
| EN | 2 | 37 | 35 | 2 |
| ENC | 2 | 37 | 10 | 27 |
| Total | 11 | 259 | 111 | 148 |

Table 6: Number of items of each label in the train/test split from each source.

---

| Instruction | Given a context, a statement can be true, false or either true or false. Different people may interpret the sentences differently and have different judgments. You are an assistant helping users identify what possible judgments people can have and provide an explanation for each judgment. You need to identify as many possible judgments as possible.
User:
**Context:** While it's probably true that democracies are unlikely to go to war unless they're attacked, sometimes they are the first to take the offensive.
**Statement:** Democracies probably won't go to war unless someone attacks them on their soil
Assistant:
Given the context, some people consider the statement to be **true**, some people consider the statement to be **either true or false**. | Explanations | **It can be true because** if a nation is being attacked, odds are it's occuring on its soil (there's far more of that than say, foreign military bases or whatever). The context states it's likely true democracies won't go to war unless they're attacked, and since the context is taken as true that means the statement must be so as well.
**It can be either true or false because** the statement adds an extra part with "on their soil" which is not explicitly mentioned in the context. This could be true or false since the context only mentions being attacked, not where the attack would take place.

Table 7: Training example prompt for the Predict-then-Explain setup. The boxed words indicate components of the prompt and do not appear in the actual prompt. The | Instruction | only appears once before all the training items. The Explain-then-Predict prompt has the | Prediction | following | Explanations |. The Predict-only prompt does not include | Explanations |.

## A4 Evaluating the Generated Explanations

Table 8 shows the definitions of the four binary criteria and examples of explanations judged according to these criteria.

Note that the sentence content partially affects the ratings for Specificity and New Information. If the label is straightforward and there is not much to be explained, a good explanation does not necessarily need to be specific. Similarly, for New Information, the sentences may not discuss any real world events/entity, and there is not much new information that can or need to be added.

Table 9 shows the evaluation results breakdown by labels.

**Hide model identity for evaluation** The identity of the model is hidden by only presenting the portion of the text that corresponds to the explanation and shuffling the order. The gold label / distribution of the item is also hidden.

| **Specificity** | **Specific / Generic** |
|---|---|

Is the explanation being detailed and specific in explaining the label, or is it more generically describing what the sentences state (or do not state) verbatim?

| **Plausibility** | **Plausible / Implausible** |
|---|---|

Is the interpretation or inferences drawn from the context/statement plausible, that is, can someone reading the context/statement possibly read them in this way? If the explanation is generically describing what the sentences state (or do not state), is the description accurate?

| **Support the label** | **Support / Not Support** |
|---|---|

Does the explanation supports the label? Does the conclusion reasonably follow the description of the premise/hypothesis? If the explanation is implausible or inaccurately described the sentences, disregard the discrepancy and assume the explanation to be true.

| **New Information** | **New info / No new info** |
|---|---|

Does the explanation introduce new facts or information about the subject matter?

---

**P:** and to have children and just get a day care or someone to take care of it and not really have the bonding process that takes place with babies and stuff you know
**H:** The day care is perfect for the children.

---

**Specific**   **Plausible**   **Support**   **New info**
N – it is not known from the context what the preference of the parent is toward day care or having another person take care of the child and the preference may vary from person to person. Furthermore, the statement does not clarify what the definition of "perfect" is for that individual.

---

**Specific**   **Plausible**   **Not Support**   **No new info**
C – taking care of children and the bonding process are two different things and one cannot infer that the day care can provide the same process as the natural bonding that takes place after the birth of a child.

---

**Specific**   **Implausible**   **Not Support**   **No new info**
C – the bonding process with babies was not mentioned in the context, suggesting that getting a day care would not be ideal for the children's needs. The statement implies that the day care is perfect, which may be false.

---

**Generic**   **Implausible**   **Support**   **No new info**
N – the context is speaking in general and does not specify that the day care is perfect. It just states that someone will take care of the children.

---

Table 8: Definitions of the four proposed criteria: Specificity, Plausibility, Support the label, and Introduce New Information, and examples of explanations judged according to these criteria.

| Setup | Label | Plausible | Support | Plausible & Support | Specific | New Info | Total |
|---|---|---|---|---|---|---|---|
| | E | 76.5 | 88.2 | 70.6 | 35.3 | 5.9 | 17 |
| E-P | N | 65.2 | 87.0 | 56.5 | 47.8 | 17.4 | 23 |
| | C | 82.1 | 71.4 | 57.1 | 32.1 | 7.1 | 28 |
| | E | 87.5 | 93.8 | 81.2 | 25.0 | 6.2 | 16 |
| P-E | N | 68.0 | 88.0 | 60.0 | 44.0 | 4.0 | 25 |
| | C | 79.3 | 69.0 | 55.2 | 37.9 | 6.9 | 29 |

Table 9: Percentage of explanations of each model with each label, marked as being Specific, Plausible, Supporting the label, and Introduce New Information. An explanation is considered to be valid if it is both plausible and supporting the label.