# OpenReview forum: "Ecologically Valid Explanations for Label Variation in NLI"
_EMNLP/2023/Conference — EMNLP 2023 Findings_

### Official Review · Reviewer_SDvm · 2023-07-30

**Soundness:** 2

**Excitement:**

2: Mediocre: This paper makes marginal contributions (vs non-contemporaneous work), so I would rather not see it in the conference.

**Justification For Ethical Concerns:**

I do not see ethical concerns with this paper. Note that the procedures in this paper passed an IRB approval.

**Missing References:**

N/A

**Paper Topic And Main Contributions:**

This paper considers the problem of human label variation in natural language inference.
First, this paper collects LiveNLI, a dataset containing 1415 annotation+explanation pairs for 122 MNLI items. This dataset confirms the label variations. Further, this dataset leads to another problem: within-label variation (annotators choose the same label for different reasons).
Then, this paper dives into the within-label variation problem. They few-shot prompt LMs to generations. The prompting experiments lead to findings including that the explanations sometimes do not support the labels.


**Questions For The Authors:**

- One could argue for an alternative perspective that the label variation problem is simply caused by some mistakes in the annotators (either those who annotated the original MNLI dataset or those recruited in this paper). Would you like to comment on this argument?

**Reasons To Accept:**

- The label variation is an important issue.
- This paper shows evidence that label variation occurs in both humans and LMs.
- This paper analyzes both the label variation and the within-label variation issues.


**Reasons To Reject:**

- Overall, this paper appears to lack a focus. The label variation and within-label variation appear like two distinct research problems, and I expect each question to be more thoroughly analyzed in a conference-level publication. These are two distinct problems, and I think analyzing one problem to a sufficient depth can constitute of a solid paper. If the central problem is to analyze the label variation, then I expect more discussions about the NLI task itself: are there naturally multiple possibilities towards the problems? Are there some mistakes in the dataset's labels? Why do the annotators write different reasons (for the within-label variation problem)? If the central problem is the within-label variation (of the models), then I expect to see some hypotheses about the models' mechanisms, some analysis to support these hypotheses, and some attempts towards mitigating the within-label variation problems.
- The problem scope is limited. While there are indeed a lot of efforts towards the NLI problem, it would be great that there is at least one other non-NLI datasets.
- A minor complaint: The term "ecologically valid" appears odd to me. Why do the annotated labels become ecologically valid when they come with explanations?

**Reproducibility:**

4: Could mostly reproduce the results, but there may be some variation because of sample variance or minor variations in their interpretation of the protocol or method.

**Reviewer Confidence:**

3: Pretty sure, but there's a chance I missed something. Although I have a good feel for this area in general, I did not carefully check the paper's details, e.g., the math, experimental design, or novelty.

**Typos Grammar Style And Presentation Improvements:**

- Line 7: MNLI needs a citation (this is the first time it occurs in this paper).

---

> ### Author Rebuttal · Authors · 2023-08-28
>
> Thank you for reviewing our paper. We were hoping to spur interest in the wider NLP community about variation in annotations, and we are glad that the reviewers recognized that aspect of our work.
>
> ### Re: lack of focus
>
> As pointed out in our introduction, this paper builds on Jiang and de Marneffe (2022): that previous work offered an in-depth analysis of the NLI task, answering many of your questions. Here we build on Jiang and de Marneffe (2022) and investigate the problem of label variation through explanations. That is, *the focus of this paper is to understand the source of label variation*, through explanations provided by the annotators themselves (i.e., ecologically valid).
>
> Within-label variation is something we discovered while analyzing label variation.  We did touch on why people can have different reasons for the same label (e.g. coreference, Question Under Discussion, line 151 - 172), and we can expand this analysis in the final version.
>
> As you pointed out, within-label variation presents its own set of questions that warrants future research. Therefore, we hope that our paper can present these questions to the NLP community and a resource to start answering those questions, or encourage others to build more resources.
>
> ### Re: the term “ecologically valid”
>
> We meant that the explanations are ecologically valid, in that they are provided by the annotators who provide the labels. In comparison, other explanations datasets only collected the explanations while taking the ground truth answers for granted (e-SNLI, Camburu et al 2015, or ECQA Aggarwal et al 2021).
>
>
> ### Re: annotator mistakes
>
> Some of the variation might indeed be due to genuine errors from the part of the annotators, as acknowledged in our Limitations section. In LiveNLI, we found 6 annotations (out of 1,415) where the explanations revealed that the annotators read the sentences incorrectly. The number of genuine mistakes is however small and label variation persists even if we discard those 6. The rest of the labels are all provided by the annotators with their best effort, as shown by the quality of their explanations. In ChaosNLI, the authors took great care into vetting the annotators, making sure that they paid attention. If label variation persists (and we show that it does), it is part of human behavior that is worth studying.

---

### Official Review · Reviewer_9znK · 2023-07-31

**Soundness:** 4

**Excitement:**

4: Strong: This paper deepens the understanding of some phenomenon or lowers the barriers to an existing research direction.

**Paper Topic And Main Contributions:**

This paper presented the LIVENLI dataset, containing 1,415 ecologically valid explanations for 122 MNLI items, to explore human label variation in natural language inference (NLI). The authors analyze label variation, including within-label variation, and find that annotators
 sometimes choose the same label for different reasons, which suggests that explanations are crucial for navigating label interpretations in
 general.  They also experiment with using LLMs to predict and explain label variation, emphasizing the need for diverse explanations to imitate human decision-making.

**Questions For The Authors:**

- In lines 40 - 42: "each with at least 10 highlights and free-text explanations for the labels chosen by the annotators, totaling 1,415 explanations", however, in lines 62 - 63: "We re-annotated 122 items from the MNLI dev set where each item has 5 annotations". How many annotations did you do exactly? I am confused here.
- I don't quite get how you use text-davinci-003 to do multilabel classification with explanations. Do you instruct the model to output multiple labels at the same time and give explanations to each label? It would be helpful if you can put the prompts you used to generate the results in the appendix.
- When using text-davinci-003, how many examples do you use as few-shot demonstrations and how stable those results are for different GPT calls? I am asking this because the test set is small and sometimes the instabilities of GPT calls can yield different conclusions.
- Did you try zero-shot prompts and compare the results with few-shot prompts?

**Reasons To Accept:**

- The introduction of the LIVENLI dataset, focusing on ecologically valid explanations, fills a gap in the field and provides a valuable resource for future research.
- The authors conducted thorough human annotation and analysis to investigate the label variations of NLI, and they find that Questions Under Discussion (QUDs) is an important factor in label variation which adds a new dimension to the taxonomy of prior work.
- The scope of this paper is well-suited for a short paper and the paper is well-presented.

**Reasons To Reject:**

- I am not quite convinced by the conclusion stated in Line 221 "This adds to the series of results where having post-prediction explanations outperforms chain-of-thought style prompting". Just like the authors mentioned in the Limitation section, it could be that the training data of text-davinci-003 already covers the test set, so that directly predicts a label achieves higher performance. I think to rule out this possibility, the authors could gather 10 or 20 new examples and test the same methods, which would make the results more convincing.
- I am a bit confused by several parts in the paper. See questions below for details.

**Reproducibility:**

4: Could mostly reproduce the results, but there may be some variation because of sample variance or minor variations in their interpretation of the protocol or method.

**Reviewer Confidence:**

4: Quite sure. I tried to check the important points carefully. It's unlikely, though conceivable, that I missed something that should affect my ratings.

---

> ### Author Rebuttal · Authors · 2023-08-28
>
> Thank you for reviewing our paper.
>
>
> ### Re: reasons to reject
>
> > I am not quite convinced by the conclusion stated in Line 221 "This adds to the series of results where having post-prediction explanations outperforms chain-of-thought style prompting"
>
> On line 221, we were mainly comparing Predict-then-Explain vs. Explain-then-Predict.
>
> > Just like the authors mentioned in the Limitation section, it could be that the training data of text-davinci-003 already covers the test set, so that directly predicts a label achieves higher performance. I think to rule out this possibility, the authors could gather 10 or 20 new examples and test the same methods, which would make the results more convincing.
>
>
> You are right that this could be a possibility. Collecting a few new examples takes more time than we have available during the author response period. Instead, we briefly tested the hypothesis of whether the model could be memorizing the MNLI labels. We zero-shot prompted 10 examples from the MNLI dev set to see if it memorizes the original labels. The prompt format is: “premise: ___ hypothesis: ___ entailment/neutral/contradiction? label:”. We specifically chose examples where only 3 out of 5 MNLI labels agreed, and they differ from LiveNLI labels. The goal is to have the original MNLI majority voted labels be more of an arbitrary choice, and if the model predicts the same label as the MNLI label, it is likely due to memorization. We found that the model predicts the same MNLI label on 5 (out of 10) examples, suggesting that even if the MNLI test set is in the training data, it does not consistently memorize the labels.
>
> On the other hand, it is possible that training on the MNLI test set leads to other advantages such as being able to better extrapolate task-specific patterns and generate better explanations. There is indeed no definite conclusion to be drawn without more information about how the model text-davinci-003 was built, which is sadly information that is not available.
>
> ### Answers to your questions
>
> - > In lines 40 - 42: "each with at least 10 highlights and free-text explanations for the labels chosen by the annotators, totaling 1,415 explanations", however, in lines 62 - 63: "We re-annotated 122 items from the MNLI dev set where each item has 5 annotations". How many annotations did you do exactly? I am confused here.
>
> We collected at least 10 annotations for each item for LiveNLI. On line 62-63, we meant that the authors of MNLI collected 5 annotations for the dev set. We will rephrase to avoid the ambiguity.
>
> - > I don't quite get how you use text-davinci-003 to do multilabel classification with explanations. Do you instruct the model to output multiple labels at the same time and give explanations to each label? It would be helpful if you can put the prompts you used to generate the results in the appendix.
>
> Here’s the prompt format – Context: ____ Statement: ___ Given the context, some people consider the statement to be _lable1_, some people consider the statement to be _label2_. It can be label1 because ___ It can be label2 because ___. We will add the prompt in the appendix in the final version.
>
> - > When using text-davinci-003, how many examples do you use as few-shot demonstrations and how stable those results are for different GPT calls? I am asking this because the test set is small and sometimes the instabilities of GPT calls can yield different conclusions.
>
> As we mentioned on line 204, we used 11 examples for few-shot demonstrations. In Table 2, we reported standard deviation of the metrics from two random splits. We can see that E-P underperforms the other two models in both runs, but the differences between P vs. P-E are small.
>
> - > Did you try zero-shot prompts and compare the results with few-shot prompts?
>
> We did not try zero-shot prompts for multilabel classification. We will consider this direction for future work.

---

### Official Review · Reviewer_sZVS · 2023-08-05

**Soundness:** 3

**Excitement:**

3: Ambivalent: It has merits (e.g., it reports state-of-the-art results, the idea is nice), but there are key weaknesses (e.g., it describes incremental work), and it can significantly benefit from another round of revision. However, I won't object to accepting it if my co-reviewers champion it.

**Missing References:**

NA

**Paper Topic And Main Contributions:**

This paper addresses the issue of human label variation in Natural Language Inference (NLI) tasks and explores how such variation arises. The authors observe that traditional practices for operationalizing annotations in NLP datasets have largely assumed a single label per item, a simplification that doesn't accurately capture the complexity of human label variation. This work explores where label variation in NLI comes from and how it can be better understood and potentially leveraged in future tasks.



**Questions For The Authors:**

See reasons to reject

**Reasons To Accept:**

- The authors' work presents new data resources, provides a new perspective on the interpretive process in NLI tasks, and conducts an NLP engineering experiment. The dataset presented can be a valuable resource for further research on NLI and the human label variation problem, providing both raw data and insights.
- Exploration of Label Variation: The paper tackles the understudied issue of human label variation in NLI, shedding light on the factors that lead to such variation. This understanding can lead to more sophisticated approaches in NLI and similar tasks, by acknowledging and incorporating label variation rather than attempting to oversimplify the problem.
- Identification of Within-Label Variation: The authors identified and highlighted a new type of variation, within-label variation, where annotators may choose the same label for different reasons. This finding adds a new layer of complexity to our understanding of label interpretation and could lead to more nuanced models and strategies in NLI tasks.


**Reasons To Reject:**

- Size of the Dataset: The LiveNLI dataset, though novel and useful, is relatively small, with only 122 re-annotated MNLI items. Its small size may limit its effectiveness in providing comprehensive insights into label variation across a wide range of contexts and applications.
- Lack of Broad Model Evaluation: The paper evaluates the ability of GPT-3 to generate explanations for label variation. However, the evaluation does not extend to other state-of-the-art models, e.g., GPT-4, which could offer a broader perspective on AI's capacity to handle such complex tasks.
- Unclear Generalizability: The findings mainly pertain to label variation in NLI tasks. While these findings are valuable, it remains unclear how broadly they apply to other NLP tasks. More discussion or analysis on the generalizability of the findings to other areas of NLP would enhance the paper's impact.

**Reproducibility:**

4: Could mostly reproduce the results, but there may be some variation because of sample variance or minor variations in their interpretation of the protocol or method.

**Reviewer Confidence:**

4: Quite sure. I tried to check the important points carefully. It's unlikely, though conceivable, that I missed something that should affect my ratings.

---

> ### Author Rebuttal · Authors · 2023-08-28
>
> Thank you for reviewing our paper. We appreciate the recognition of human label variation as an important problem. We would like to point out that **our main contribution lies in the collection of ecologically valid explanations of label variation**. We were hoping to spur interest in the wider NLP community about variation in annotations, and we are glad that the reviewers recognized that aspect of our work.
>
>
> ### Re: Size of the dataset and Lack of Broad Model Evaluation
> We understand the limitations of this work being narrowly scoped, focusing on one task and collecting a small dataset. We intended our work to be a focused contribution, a proof of concept, and an initial exploration in this topic as a short paper. Even though the number of items (122) is small, we collected a large number of explanations for each item (10+, most explanation datasets contain only 3 or 5 explanations and sometimes even just one.). We performed manual examinations on all the explanations, as automatic methods that we experimented with did not reveal much insights. We only tested one model due to space constraints, and we did not have access to the more powerful GPT-4 at the time. We will address these limitations in the Limitations section in our revision.
>
>
> ### Re: Unclear Generalizability
>
> Even though we only analyzed the NLI task, our findings can inform approaches to other tasks such as fact-checking. For example, our findings suggest that people’s judgments vary because they consider different Questions Under Discussion(QUD), which in turn suggests that variation can be reduced if we make QUDs explicit. This is important in the task of fact-checking, where all the QUDs need to be considered and a claim is only true if all of its possible implications, i.e. answers to all QUDs, are true. [Chen et al 2022](https://aclanthology.org/2022.emnlp-main.229/)) presents a step in this direction. They introduced a dataset/method for fact-checking complex claims where each claim is accompanied by a set of questions (that can be considered as QUDs), and a claim is fact-checked by checking answers to all these questions. By making questions explicit, it can reduce potential variation and facilitate checking the entire claim.

---

### Meta-Review · Area_Chair_GUpy · 2023-09-14

**Recommendation:** 5

**Metareview:**

The paper addresses an interesting problem, one of the common objections is the limited scope and the small size of the provided resource. However, considering that it is a short paper, the scope and the contribution of the paper are adequate.

---

### Decision · Program_Chairs · 2023-10-07

**Decision:**

Accept-Findings

**Comment:**

The paper addresses an interesting problem, one of the common objections is the limited scope and the small size of the provided resource. However, considering that it is a short paper, the scope and the contribution of the paper are adequate.